# Effect of Oats and Wheat Genotype on In Vitro Gas Production Kinetics of Straw

**DOI:** 10.3390/ani11061552

**Published:** 2021-05-26

**Authors:** Karen A. Peñailillo, María Fernanda Aedo, María Carolina Scorcione, Mónica L. Mathias, Claudio Jobet, Manuel Vial, Iris A. Lobos, Rodolfo C. Saldaña, Paul Escobar-Bahamondes, Paulina Etcheverría, Emilio M. Ungerfeld

**Affiliations:** 1Departamento de Ingeniería Química, Facultad de Ingeniería y Ciencias, Universidad de la Frontera, Temuco 4780000, La Araucanía, Chile; k.penailillo01@gmail.com; 2Centro Regional de Investigación Carillanca, Instituto de Investigaciones Agropecuarias INIA, Vilcún 4880000, La Araucanía, Chile; aedo.enriquez13@gmail.com (M.F.A.); monica.mathias@inia.cl (M.L.M.); cjobet@inia.cl (C.J.); manuel.vial@inia.cl (M.V.); paul.escobar@inia.cl (P.E.-B.); paulina.etcheverria@inia.cl (P.E.); 3Facultad de Agronomía, Universidad de Buenos Aires, Buenos Aires C1417DSE, Provincia de Buenos Aires, Argentina; caroscorcione@gmail.com; 4Centro Regional de Investigación Remehue, Instituto de Investigaciones Agropecuarias INIA, Osorno 5290000, Los Lagos, Chile; iris.lobos@inia.cl (I.A.L.); rsaldana@inia.cl (R.C.S.)

**Keywords:** straw, ruminants, genotypes, oats, wheat, gas production, in vitro

## Abstract

**Simple Summary:**

Increases in cereal grain yields cause the accumulation of large amounts of straw on the soils after grain harvest. Straw is usually burned in the field to help soil preparation for the next crop, a practice resulting in local and global pollution, erosion, loss of soil carbon, and wildfires. An alternative is feeding straw to ruminants, but straw has poor nutritive value, making this option unattractive to Chilean farmers. Oats and wheat have been bred for greater grain yield and improved agronomic traits, but it is unknown whether the straw of different varieties and breeding lines differs in nutritive quality. To investigate this possibility, we incubated the straws from 49 different varieties and breeding lines of oats and 24 of wheat with rumen microorganisms, and studied gas production as an indication of the extent of straw digestion. We found moderate differences among varieties and breeding lines of oats and wheat in gas production, which were not detrimental to agronomic characteristics of importance. If these results can be confirmed in animal experiments, gas production of straw incubated in rumen microbial cultures may be used to identify cereal genotypes whose straw has a better nutritive quality for ruminants.

**Abstract:**

Increases in cereals grain yield in the last decades have increased the accumulation of straw on the soil after harvest. Farmers typically open burn the straw to prepare the soil for the next crop, resulting in pollution, emission of greenhouse gases, erosion, loss of soil organic matter, and wildfires. An alternative is feeding straw to ruminants, but straw nutritive value is limited by its high content of lignocellulose and low content of protein. Cereal breeding programs have focused on improving grain yield and quality and agronomic traits, but little attention has been paid to straw nutritive value. We screened straw from 49 genotypes of oats and 24 genotypes of wheat from three cereal breeding trials conducted in Chile for in vitro gas production kinetics. We found moderate effects of the genotype on gas production at 8, 24, and 40 h of incubation, and on the maximum extent and rate of gas production. Gas production was negatively associated with lignin and cellulose contents and not negatively associated with grain yield and resistance to diseases and lodging. Effects observed in vitro need to be confirmed in animal experiments before gas production kinetics can be adopted to identify cereal genotypes with more digestible straw.

## 1. Introduction

Cereal yields have nearly duplicated in Chile between the 1980/1989 and 2000/2009 decades [1], resulting in sustained increases in the production of crop residues [2]. The average accumulation of straw on the soil in Chile after cereal harvest has been estimated as 6.4, 7.6, 6.9, 14.2, and 7.7 ton/ha for wheat, oats, barley, corn, and rice, respectively, of which only between 2.5 and 3 ton/ha are estimated to decompose every year [3].

Producers frequently burn the straw on the field to help soil preparation for the next crop [4]. Open burning of straw is legally regulated in Chile by Ordinance 276/1980 of the Ministry of Agriculture [5], and it is the most widely used practice to manage straw residues. However, this practice is highly undesirable because it oxidizes soil organic matter (OM), produces greenhouse gases, favors soil erosion and compaction, pollutes the air, and causes wildfires [6,7,8,9].

An alternative to burning straw is to feed it to ruminants. Feeding crop residues to livestock is resource-efficient as it allows integrating ruminant and crop production by recycling nutrients to the soil as animal manure and supplying traction for soil preparation and sowing [10]. Mixed crop–livestock systems also contribute to spreading financial risks and increasing food security of smallholders by using a very low-cost feed [11]. However, even though ruminants have the ability to digest fiber, cereal straws have a low nutritive value because their high content of lignocellulose and low nitrogen content pose constraints on digestibility and voluntary intake [2,12]. In Chile, Instituto de Investigaciones Agropecuarias INIA started breeding wheat and oats in the 1960s with a focus on grain yield, industrial quality, and disease and lodging resistance. However, how cereal breeding efforts might affect the composition and nutritional quality of straw for ruminants has not been investigated in Chile, despite the growing importance of problems caused by accumulation and open burning of straw. Previous research has reported variation in nutritional quality of residues among cultivars of wheat, barley, oats, and sorghum [2,13,14,15]. It would be important to identify those cereal varieties and breeding lines with a straw with superior nutritive value for ruminants. Equally important, cereal genotypes possessing a nutritionally superior straw should also maintain desirable agronomic characteristics, such as high grain yield and resistance to diseases and lodging.

Rumen digestion of large numbers of plant materials can be preliminarily screened through the use of in vitro gas production techniques [16,17]. We hypothesized the existence of differences in gas production in vitro of straw from different wheat and oats genotypes. We also hypothesized that the nutritive value of straw as assessed through in vitro gas production would not associate unfavorably with grain yield and incidence and severity of lodging and diseases. The objectives of this research were (1) to evaluate the variation in gas production kinetics of straw from different genotypes of wheat and oats grown and bred in Chile, and (2) to investigate the relationship between in vitro gas production and agronomic traits.

## 2. Materials and Methods

### 2.1. Cereal Breeding Trials

Three breeding trials for evaluating advanced breeding lines and commercial varieties of oats and wheat were sown by the cereal breeding programs of Instituto de Investigaciones Agropecuarias INIA at Centro Regional Carillanca, Vilcún, La Araucanía, Chile (38.69° S, 72.41° W; 200 m above sea level). The Oats 1 trial, sown in June 2018, included 5 commercial varieties and 20 advanced breeding lines. The Oats 2 trial, sown in July 2018, included 2 commercial varieties currently in the market (registered in 2007 and 2015), 22 historical varieties (registered between 1968 and 2004), and a landrace genotype. The Wheat trial, sown in June 2018, included 4 commercial varieties and 21 advanced breeding lines.

All three genotype evaluation trials were sown as random block designs, with four blocks per genotype. Within each trial and block, oats or wheat genotypes were randomly assigned to 2 × 1 m experimental plots and hand-planted at 120 kg/ha of seed and 0.2 m of separation between rows. Crops were fertilized with 180 kg/ha N (20% at sowing, 40% at the beginning of tillering, and 40% at full tillering), 80 kg P_2_O_5_ (100% at sowing) and 70 kg/ha K (100% at sowing). Weeds were controlled using a post-emergence herbicide mixture of 10 g/ha metsulfuron methyl (2-[4-methoxy-6-methyl-1,3,5-triazin-2-ylcarbamoylsulfamoyl]benzoic acid; Anasac Chile S.A., Santiago, Chile), 0.5 L/ha MCPA-dimethylammonium (A.H. Marks and Company Ltd, West Yorkshire, UK) and 170 g/ha dicamba-sodium (Syngenta Crop Protection AG, Basel, Switzerland), diluted in 200 L/ha water. Crops were grown without application of fungicides, insecticides, or growth regulators and were harvested with a combine harvester machine. At harvest, grain yield was recorded for each experimental plot from a clean sample of grain and adjusted to 12% moisture content.

In both oats breeding trials, plant height at harvest was recorded as the sum of stem and panicle. Plant lodging was recorded in both oats trials right before harvest both in terms of incidence (visual percentage of the experimental plot area) and severity (on a 1 to 5 subjective scale with 1 being the absence of tilting and 5 being complete inclination). In both oats trials, the incidence of Halo blight (*Pseudomona syringae* pv. *coronafaciens*) was estimated as the visual percentage of damaged leaves, and the incidence of Barley yellow dwarf as a percentage of affected plants in the plot was estimated visually. In the Oats 1 trial, the incidence of crown rust (*Puccinia coronata* var. *avenae* f. sp. *avenae*) was evaluated with the modified binomial system of Cobbs [18]. All disease scores were recorded three times during the oat crop cycle: when the first node of the stem became visible, at the end of flowering, and during dough grain. In each experimental plot, the maximum disease score was considered for the analyses of correlation with gas production parameters (see Section 2.5).

### 2.2. Straw Collection, Morphological Measurements and Proximate Analyses

Straw residues were collected from 1 m of the central row of each experimental plot in both oats trials and the wheat trial (*N* = 100 per crop breeding trial corresponding to 25 genotypes and 4 blocks) no later than one week after grain was harvested. All of the straw samples from each of the breeding trials were collected within one day. Residues were cut at the ground level and the resulting straw was then cut from the top to 40 cm long. This resulted in samples of approximately 200 g straw. The collected samples did not contain chaff.

From each sample of straw, 10 representative tillers were subsampled, individually weighed, and used for morphological determinations. The leaves (sheaths and blades together) were separated from the stem of each tiller. The leaves and stems were then weighed separately, and the mass percentage of leaves (*%Leaves*) calculated as g⁄100 g sample. The diameters of the first and second internodes of each tiller were measured using a micrometer. The ratio of the first to the second internode diameter (*R**Φ*) and the average diameter of the first and second internodes (*av**Φ*) were calculated and expressed in mm⁄mm and mm, respectively. The average diameter was used to estimate the area section assuming a circular shape, and the straw volume estimated by multiplying the so obtained area section by the straw length assuming a cylindrical shape of the stem. Finally, the mass of each stem was divided by its estimated volume to obtain the stem apparent density (*δ*), expressed in g⁄cm^3^.

Each subsample of the 10 tillers used for morphological determinations was then returned to its original sample. From each sample, approximately 40 g were again subsampled, and the four 40 g subsamples from each of the four blocks corresponding to each genotype were combined into a pooled sample per genotype (*N* = 25 per crop breeding trial). The resulting pooled samples were shipped to Laboratorio de Nutrición y Medio Ambiente, Instituto de Investigaciones Agropecuarias, Centro Regional Remehue, Osorno, Chile, for their proximate compositional and in vitro digestibility analyses. Pooled samples were ground through a 1 mm screen and analyzed for dry matter (DM, method 934.01), total ash (method 942.05), crude protein (CP), method 984.13) [19], and fiber fractions neutral detergent fiber (NDF), acid detergent fiber (ADF) and acid detergent lignin (ADL) excluding residual ash [20], as well as apparent in vitro DM digestibility (IVDMD) [21]. Hemicellulose and cellulose contents were estimated by subtracting ADF from NDF and ADL from ADF, respectively [20,22].

### 2.3. In Vitro Incubations

The ground, combined samples from each wheat and oats genotype used for the analyses of proximal composition were incubated in in vitro mixed rumen cultures at Instituto de Investigaciones Agropecuarias INIA at Centro Regional Carillanca, Vilcún, La Araucanía, Chile. Genotypes were incubated in separate experiments for each breeding trial Oats 1, Oats 2 and Wheat. One genotype of each of the Oats 1 and the Wheat trials were not incubated because they were temporarily lost after their proximate analyses and hence their shipping to the incubation laboratory was delayed.

Rumen contents were sampled before the morning feeding from two ruminally cannulated, non-lactating, non-pregnant, Holstein cows ad libitum fed wheat straw (90.2% DM, and on a DM basis 6.30% ash, 2.00% CP, 79.5% NDF, 54.0% ADF, and 4.51% ADL) and a mineral block (Veterblock, Veterquímica S.A., Santiago, Chile). Rumen contents were strained through a double synthetic cloth, and the resulting fluid and solid fractions immediately transported to the laboratory in separate insulated containers.

All rumen inoculum preparation procedures in the laboratory were conducted under O_2_-free CO_2_. Two hundred milliliters of pooled rumen fluid from both cows were combined with 100 mL of solids from both animals, and blended at low speed for 1 min discontinuously (3 s blending followed by 2 s interruptions) to detach microbial cells adhered to solid particles. The beaten rumen contents were strained through two synthetic layers and the procedure was repeated until obtaining 600 mL of rumen fluid enriched with detached microbial cells. The resulting 600 mL of rumen fluid were combined with the medium of Mould et al. [23] in a 1:4 ratio (*V/V*) [24]. Of the resulting inoculum, 40 mL were delivered into 100 mL serum bottles containing 501 ± 1.37 (mean ± SD) mg of ground straw substrate of a particular genotype. Rumen inoculum was constantly agitated with a rotating magnet while being delivered into the serum bottles. Bottles were crimp sealed with butyl rubber stoppers under O_2_-free CO_2_, and incubated in a shaking water bath at 39 °C and 60 rpm for 72 h.

Gas pressure accumulating in each bottle was measured at 5 min and 2, 3, 6, 9, 12, 18, 24, 36, 48, and 72 h from the beginning of the incubation using a pressure transducer (Sper Scientific 840065, Scottsdale, AZ, USA), without allowing for gas release [24] and without removing the bottles from the water bath at 39 °C. The exact time at which each measurement was conducted was recorded for each bottle and time point. After the last measurement of gas pressure at 72 h of incubation, bottles were opened and pH immediately measured (Oakton^®^ pH 700 m, Vernon Hills, IL, USA).

Each genotype was incubated in duplicate. Each of the duplicate bottles of each genotype was randomly allocated to one of two groups blocked by order of inoculation. For each crop breeding trial, four incubations were conducted in different weeks, totaling 200 bottles in the Oats 2 breeding trial (25 genotypes × 2 incubation duplicates × 4 incubations) and 192 bottles in the Oats 1 and Wheat breeding trials (24 genotypes × 2 incubation duplicates × 4 incubations).

### 2.4. Calculations

For each serum bottle, accumulated gas pressure was modelled as an inverted exponential function as a function of time elapsed from the beginning to the incubation in that bottle [17], using the exact time points at which gas pressure was measured in each bottle:*P* = *a* + *b* (1 − exp^−*ct*^)(1)
where *P* is accumulated gas pressure expressed in atm at time *t* in h, *a* is the intercept, and *b* is the theoretical maximum increment in gas pressure, both in atm, and *c* is the fractional rate of increase in gas pressure in h^−1^ [17]. The theoretical maximum accumulated gas pressure was calculated as the sum of *a* and *b*:(2)$$\underset{t\;\to +\infty }{\lim}P=\underset{t\;\to +\infty }{\lim}\left(a+b\;\left(1-\exp(-ct\right)\right))=a+b$$

The parameterized gas pressure equations for each incubated bottle were also used to estimate predicted accumulated gas pressure at 8, 24, and 40 h of incubation. Gas production in mmol at 8 (*P*_8_), 24 (*P*_24_) and 40 (*P*_40_) h and the theoretical maximum gas production (*P_max_*) were calculated from gas accumulated at 8, 24, and 40 h, and from the theoretical maximum accumulated gas pressure, respectively, using the ideal gas law considering a 0.060 L gas headspace and a temperature of 312 K. Gas pressure parameters *a* and *b* were also used to calculate gas production intercept *a* and maximum increment *b* using the ideal gas law.

### 2.5. Statistical Analyses

All response variables were separately analyzed per breeding trial. The genotype was modelled as a random variable in order to extrapolate the results from the populations of genotypes examined to conceptually larger populations of genotypes beyond the genotypes evaluated in each trial.

For each breeding trial, the random effect of the genotype on morphological response variables *avΦ*, *RΦ*, %leaves, and δ was separately modelled for trials Oats 1, Oats 2, and Wheat as:(3)response=overall mean+block+genotype(random)+error
where block corresponds to the fixed effect of the location of each experimental plot within each of the breeding trials Oats 1, Oats 2 and Wheat, from which each straw sample was collected.

The random effect of the genotype on the rumen incubation parameters *a*, *b*, *c*, *P*_8_, *P*_24_, *P*_40_ and *P_max_*, and final pH, was separately modelled for trials Oats 1, Oats 2, and Wheat as:(4)response=overall mean+inoculation+genotype(random)+incubation(random)+error
where the fixed effect of inoculation refers to the order in which each duplicate belonging to each genotype was inoculated (first or second batch), and incubation refers to the random effect of the four incubation runs conducted on separate weeks.

All two-way associations between incubation parameters *a*, *b*, *c*, *P*_8_, *P*_24_, *P*_40_*, P_max_*, and final pH were examined through Pearson correlations using the results of all incubation bottles. The associations between *P*_8_, *P*_24_, *P*_40_*, P_max_* and *c* with proximate composition (DM and OM, CP, cellulose, hemicellulose, ADL percentage in the DM) and morphological variables (*avΦ*, *RΦ*, % leaves, and δ) per genotype were studied, building multiple regression models with the backwards stepwise regression procedure removing from the models regressors with *p* > 0.05, until obtaining the final model for each response variable (*N* = 73; 25 genotypes of the Oats 2 breeding trial and 24 genotypes of each of the Oats 1 and Wheat breeding trials). Single regressions between IVDMD and *P*_8_, *P*_24_, *P*_40_*, P_max_* and *c* were also conducted.

The relationships between the gas production parameters *P*_8_, *P*_24_, *P*_40_*, P_max_* and *c* with agronomic variables were studied separately for each breeding trial (*N* = 24 for the Oats 1 and Wheat breeding trials and *N* = 25 for the Oats 2 breeding trial) through Pearson correlations. The agronomic variables examined were grain yield (Oats 1, Oats 2, and Wheat breeding trials), lodging percentage and severity, and incidence of Halo blight and Barley yellow dwarf (Oats 1 and Oats 2 breeding trials) and crown rust (Oats 1 trial). In the Oats 2 historical variety trial, *P*_8_, *P*_24_, *P*_40_*, P_max_* and *c* were regressed against the year of registration of the varieties.

Outliers were identified as observations with studentized residuals > |*t_N_*_−1,0.95_|, with *N* being the number of observations. Influential observations were identified as those with a Cook’s distance > *F*_0.5(*p*,*N*−*p*),_ with *p* being the number of parameters in the regression equation [25].

All statistical analyses were conducted with JMP^®^ 13.2.1 [26].

## 3. Results

### 3.1. Chemical Composition

Means, standard deviation, and ranges per trial for the contents of DM, OM, CP, NDF, ADF, ADL, and IVDMD of the straw residues by trial are provided in Table 1. Contents of DM and OM varied little among cereal genotypes, while there was more variation in CP, ADL, and IVDMD, with NDF, ADF, hemicellulose, and cellulose being intermediate.

### 3.2. Straw Morphology

Genotype affected the straw residue *av**Φ* (*p* ≤ 0.002), *Leaves%* (*p* ≤ 0.004), and *δ* (*p* ≤ 0.004), and affected (*p* ≤ 0.049) or tended (*p* = 0.060) to affect *R**Φ* (Appendix A).

### 3.3. In Vitro Incubations

Incubation bottles with noticeable gas leaks or those ones whose gas pressure curve against time that did not adjust to an inverted exponential function were discarded, after which 169 bottles remained for analysis in the Oats 1 and Wheat trials and 183 in the Oats 2 trial.

Plant genotype did not affect gas production intercept *a* in the Oats 1 (*p* = 0.16; Table 2) and Oats 2 trials (*p* = 0.53), but influenced it in the Wheat trial (*p* = 0.025). Gas production maximum increment *b* was affected by plant genotype in all three breeding trials (*p* ≤ 0.020). The rate of gas production *c* was affected by plant genotype in the Oats 2 and Wheat trials (*p* ≤ 0.021), and tended (*p* = 0.056) to be affected by plant genotype in the Oats 1 trial. Gas production at 8 h in the Oats 2 trial was affected by plant genotype (*p* = 0.024), and tended (*p* ≤ 0.084) to be affected by plant genotype in the Oats 1 and Wheat trials. In all three breeding trials, plant genotype affected *P*_24_ (*p* ≤ 0.018; Figure 1), *P*_40_ (*p* ≤ 0.020), *P_max_* (*p* ≤ 0.034), and final pH (*p* ≤ 0.007).

Gas parameter *a* was negatively associated with *b* and *c* (*p* < 0.001) and positively associated with *P*_8_, *P*_24_, *P*_40_, *P_max_*, and pH (*p* < 0.001; Table 3). Gas production parameter *b* was negatively associated with *c* and pH (*p* < 0.001) and positively associated with *P*_8_, *P*_24_, *P*_40_, and *P_max_* (*p* < 0.001). Gas production parameter *c* was positively associated with gas production at time points *P*_8_ and *P*_24_ (*p* < 0.001) and negatively correlated with *P*_40_ (*p* < 0.05) and *P_max_* (*p* < 0.001). Gas production at different time points was positively correlated with each other (*p* < 0.001). Final pH was negatively correlated with gas production parameter *b* and with *P*_24_, *P*_40_, and *P_max_* (*p* < 0.001), and positively with gas production parameter *a* (*p* < 0.001).

### 3.4. Associations of Gas Production Parameters and pH with Chemical Composition and Straw Morphology

Gas production at 8, 24, and 40 h of incubation was negatively related to ADL and cellulose (*p* ≤ 0.006; Table 4). The theoretical maximum gas production was negatively related to ADL only (*p* < 0.001). Gas pressure at 8 h of incubation was positively associated with *avΦ* (*p* < 0.001), whereas the fractional rate of gas production *c* was negatively associated with *RΦ* (*p* = 0.002). In vitro apparent digestibility of DM was positively related to *P*_8_, *P*_24_, *P*_40_ and *P_max_* (R^2^ ≥ 0.39; *p* < 0.001) and negatively related to *c* (*R*^2^ = 0.10; *p* = 0.006; Appendix A).

### 3.5. Association between In Vitro Gas Production and Agronomic Traits

Gas pressure at 8 h of incubation was unrelated to grain yield in the Oats 1 and Oats 2 (*p* ≥ 0.16) breeding trials, and tended (*p* = 0.076) to be associated positively with grain yield in the Wheat trial (Appendix A). In all three trials, *P*_24,_
*P*_40_, *P_max_* and *c* were unrelated to grain yield (*p* ≥ 0.12). Lodging incidence and severity were negatively related to *P*_8_ (*p* ≤ 0.024) in the Oats 1 breeding trial. Lodging incidence tended to be negatively related to *P*_24_ and *P*_40_ (*p* ≤ 0.070) and lodging severity to *c* (*p* ≤ 0.096) in the Oats 2 breeding trial. Lodging incidence and severity were otherwise unrelated to gas production in both trials (*p* ≥ 0.12).

There were no relationships between *P*_8_, *P*_24_, *P*_40_, *P_max_* and *c* with the incidence of Halo blight or *Barley yellow dwarf virus* (*p* ≥ 0.29; Appendix A) in the Oats 1 and Oats 2 breeding trials, and with the incidence of crown rust in the Oats 1 trial (*p* ≥ 0.20).

There was no relationship between the year of registration of oats varieties in the Oats 2 breeding trial and *P*_8_, *P*_24_, *P*_40_, *P_max_* and *c* (*p* ≥ 0.17) (results not shown).

## 4. Discussion

It has been proposed that variation in the nutritive value of straw could be incorporated into crops breeding programs as a trait for genetic selection [11,13,27]. This proposition requires the existence of genetic variation in the nutritive value of straw, and the absence of undesirable associations between the nutritive value of straw and those agronomic traits important to crop production [27].

In the present study, we compared in vitro gas production of straw of various genotypes of wheat and oats at 8, 24 and 40 h of incubation, as well as the theoretical maximal gas production *P_max_*, and the fractional rate of gas production. Those time points were chosen because of their relationship with different animal intake and digestion variables. Rumen retention times of around 40 h were reported for diets based on wheat [28] and barley [29,30] straw; hence we evaluated the effect of the genotype on gas production adjusted to what could be a typical rumen retention time for straw. We found that straw of different genotypes of wheat and oats differed in gas production at 40 h of incubation, although the differences were moderate, with the maximum *P*_40_ being between 22 (Oats 2 trial) and 12% (Wheat trial) over the minimum *P*_40_, and between 9.9 (Oats 2 trial) and 4.7% (Wheat trial) over the median *P*_40_.

Gas production at 8 h of incubation was a better predictor of DM intake (DMI) of cattle and sheep fed straws from 54 varieties of wheat and barley than gas production at earlier or later incubation time points [17,31]. Thus, we also evaluated gas production adjusted to the earlier incubation time point of 8 h, finding significant effects of the genotype (Oats 2 trial) or tendencies (Oats 1 and Wheat trials).

While gas production at 8 h has been shown to be the best predictor of DMI, gas production at early time points can be highly influenced by the ratio of volume of rumen inoculum to the amount of substrate [31] and day-to-day changes in the inoculum fibrolytic activity [24], in comparison with gas production at later time points. Thus, we also studied gas production at an intermediate time point of 24 h, which was regarded as being more stable among incubations and experiments compared to the early 8 h time point, and, unlike later time points, was at the same time positively related to gas production rate *c*. The effect of the crop genotype on *P*_24_ was also significant in all three trials—although the same as with other time points, the extents of the differences were moderate. In any case, *P*_8_, *P*_24_, *P*_40_, and *P_max_* were all positively and highly correlated with each other, which leads us to interpret that the main conclusions about the effect of wheat and oats genotype on straw digestion hold for all gas production time points.

Correlations between gas production kinetic parameters *a*, *b*, and *c* imply that information provided by one parameter is partially contained in others [31]; however, although all three parameters *a*, *b*, and *c* were significantly and negatively correlated in our study, correlations were numerically lower compared to the study by Blümmel and Becker [31]. Gas production rate *c* and *P_max_* were moderately and positively associated in the study by Blümmel and Becker [31]. In contrast, we found a negative association between gas production rate *c* and *P_max_*, which is undesirable as it implies that more rapid digestion of the potentially digestible fraction is associated with a greater proportion of undigestible fractions. We found moderate effects of oats and wheat genotypes on gas production kinetics parameters. Genetic effects have been reported for proximate composition, in vitro digestibility, and gas production of straw of wheat, barley, and oats [14,32,33,34,35,36], with variable results in other studies [37,38]. Previous work also reported differences among sorghum cultivars in stover composition and in vitro [15,39,40] and in vivo OM digestibility, and intake, and nitrogen balance [15,41]. Differences among historical oats varieties (Oats 2 trial) existed but did not relate to the year of registration, which might be expected as straw compositional and nutritional characteristics were not introduced as a criterion for genetic selection at any point in time.

Although not statistically analyzed, straws of different genotypes of wheat and oats also ranged considerably in CP and ADL contents, and in IVDMD. Cereal varieties can differ widely in straw composition [2], which affects straw nutritive value. Variation in in vitro digestibility of oats and wheat [14,34,42] and barley [37] straw has also been reported. Other works reported less variation in in vitro digestibility of wheat straw [35,36,43].

We included cellulose and hemicellulose as regressors available for the stepwise multiple regression procedures rather than the originally determined NDF and ADF fractions, because the correlations between cellulose, hemicellulose, and ADL were lesser than the correlations between NDF, ADF and lignin, as the sequential fiber analysis causes ADF to be contained in NDF and ADL to be contained in ADF [20]. It was not surprising that ADL had a consistently negative association with gas production, as lignin constitutes a barrier to complete digestion of cell wall polysaccharides [44,45]. The positive association between rate of gas production *c* and lignin content was likely the result of the negative association between the rate and the maximum gas production *P_max_*, as *P_max_* was in turn negatively related to lignin content. Cellulose content was also negatively related to gas production, except for *P_max_*, indicating perhaps that most cellulose was potentially but slowly digestible. Cellulose is digested in the rumen primarily by specialized bacteria and fungi, but the architecture of the plant cell wall and the interactions between cellulose and lignin and hemicellulose in the cell wall limit the rate of cellulose digestion [46,47,48].

There were significant effects of the genotype of wheat and oats on the morphological variables of straw measured. In agreement with our results, considerable variation in the percentage of leaves and other botanical fractions in crop residues of wheat, oats and barley has been reported [2]. Stems are less digestible than leaves [14,37,49], although in our study the percentage of leaves was not selected as an important explanatory variable of gas production in the stepwise multiple regressions. Conversely, stem average diameter was associated positively with *P*_8_, and the ratio between the diameters of the first and second internodes was associated negatively with the fractional rate of gas production. The stem diameter might then be associated with histological characteristics influencing the early stages of microbial digestion, hence perhaps impacting *P*_8_ and *c*. A relationship between barley stem diameter and in vitro digestibility was not previously found [13], although in that research in vitro digestibility was determined through the Tilley and Terry [50] and the Goto and Minson [51] pepsin-cellulase methods, both of which extend beyond the early digestion period.

In agreement with the present study, previous research reported no undesirable associations between the nutritive value of crop residues of wheat, barley, oats, rye, and triticale evaluated in vitro or in sacco, and agronomic traits such as grain yield and incidence of diseases and lodging [33,49,52,53,54]. On the other hand, weak but negative associations were found more recently between grain yield and straw quality traits in wheat [35,36]. Colucci et al. [32] found no relationships between straw OM in sacco digestibility and grain yield in wheat and oats, but a negative relationship in barley. There was no relationship between intake of digestible OM intake of sorghum stovers by steers and sorghum grain yield [41].

In agreement with other studies, variation in nutritional quality of straw was moderate [35,36]. However, the differences observed could potentially translate into economic differences in straw price. In Indian markets, good-quality wheat straw can be paid between 10% and 17% more than bad-quality straw [35]. A moderate increase in in vitro digestibility of sorghum stovers of five percentage units was associated with an increase of 28% in market price [55]. Although not statistically analyzed, the range in IVDMD observed in our three breeding trials was considerable, implying the possibility of the quality of straw influencing its price.

## 5. Conclusions

Moderate differences in gas production kinetics were found among genotypes of oats and wheat. Gas production was not associated undesirably with grain yield in oats and wheat and with resistance to diseases and lodging in oats. Future research should compare in situ and in vivo digestibility of genotypes contrasting in in vitro gas digestion kinetics identified in the present study. If experiments with animals confirm the present results, gas production kinetics can be used to identify breeding lines and varieties of oats and wheat with greater digestibility.

## Figures and Tables

**Figure 1 animals-11-01552-f001:**
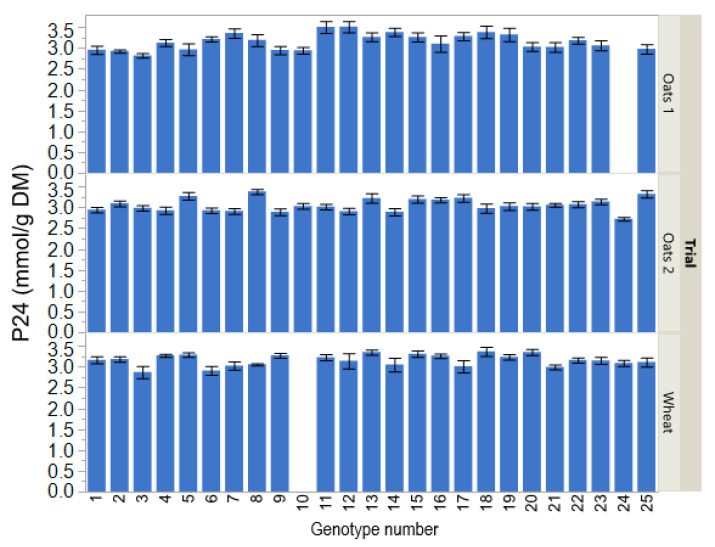
Gas production at 24 h of incubation (*P*_24_) of straw from different oats and wheat genotypes from three breeding trials (Oats 1, advanced breeding lines and commercial varieties; Oats 2, commercial and historical varieties; Wheat, advanced breeding lines and commercial varieties) in in vitro rumen cultures. Each error bar is constructed using one standard error of the mean.

**Table 1 animals-11-01552-t001:** Descriptive statistics of proximate composition of straw residues of 50 oats and 25 wheat genotypes from three breeding trials (Oats 1, advanced breeding lines and commercial varieties; Oats 2, commercial and historical varieties; Wheat, advanced breeding lines and commercial varieties).

Breeding Trial	Fraction	Mean	SD ^1^	Min	Max
Oats 1	DM ^2^ (%)	90.6	0.89	89.9	94.7
	OM (%DM)	93.4	0.80	92.2	95.6
	CP (%DM)	2.28	0.30	1.81	3.00
	NDF (%DM)	79.1	1.69	75.3	81.8
	ADF (%DM)	53.3	2.07	49.6	58.3
	ADL (%DM)	5.63	0.91	4.34	7.94
	Hemicellulose (%DM) ^3^	25.8	1.67	22.0	28.8
	Cellulose (%DM) ^4^	47.7	1.56	45.1	50.3
	IVDMD	50.4	5.43	32.9	58.6
Oats 2	DM (%)	90.6	0.34	90.2	91.4
	OM (%DM)	93.6	0.61	92.7	95.1
	CP (%DM)	2.21	0.27	1.69	2.88
	NDF (%DM)	80.5	2.24	78.1	89.7
	ADF (%DM)	55.6	1.69	52.0	59.8
	ADL (%DM)	6.49	0.77	4.87	8.25
	Hemicellulose (%DM)	25.0	2.51	22.5	35.6
	Cellulose (%DM)	49.1	1.27	46.8	51.7
	IVDMD	44.6	4.88	35.8	52.3
Wheat	DM (%)	90.3	0.29	89.8	91.0
	OM (%DM)	91.5	0.94	89.7	93.1
	CP (%DM)	2.86	0.42	1.88	3.81
	NDF (%DM)	77.2	1.62	74.1	80.3
	ADF (%DM)	52.0	1.60	49.5	55.6
	ADL (%DM)	5.76	0.71	4.37	7.22
	Hemicellulose (%DM)	25.2	1.43	22.5	27.5
	Cellulose (%DM)	46.2	1.20	43.2	48.4
	IVDMD	48.7	3.01	41.2	53.2

^1^ SD: standard deviation; Min: minimum; Max: maximum. ^2^ DM: dry matter; OM: organic matter; CP: crude protein; NDF: neutral detergent fiber; ADF: acid detergent fiber; ADL: acid detergent lignin; IVDMD: in vitro dry matter digestibility. ^3^ Calculated as NDF−ADF [20,22]. ^4^ Calculated as ADF−ADL [20,22].

**Table 2 animals-11-01552-t002:** Effect of the genotype of oats and wheat from three breeding trials (Oats 1, advanced breeding lines and commercial varieties; Oats 2, commercial and historical varieties; Wheat, advanced breeding lines and commercial varieties) on gas production parameters and final pH of straw incubated in in vitro rumen cultures.

Breeding Trial	Response	Overall Mean	SEM ^1^	Min	Max	Genotype *p* =
Oats 1	*a* (mmol/g DM incubated) ^2^	0.789	0.128	0.729	0.848	0.16
	*b* (mmol/g DM incubated)	3.53	0.129	3.20	3.99	0.008
	*c* (h^−1^)	0.048	0.004	0.043	0.053	0.056
	*P*_8_ (mmol/g DM incubated)	1.90	0.0995	1.84	2.02	0.084
	*P*_24_ (mmol/g DM incubated)	3.16	0.130	2.96	3.41	0.015
	*P*_40_ (mmol/g DM incubated)	3.75	0.146	3.50	4.10	0.011
	*P_max_* (mmol/g DM incubated)	4.32	0.183	4.05	4.73	0.017
	Final pH	6.26	0.026	6.20	6.35	0.007
Oats 2	*a* (mmol/g DM incubated)	0.506	0.0378	0.497	0.527	0.53
	*b* (mmol/g DM incubated)	3.51	0.103	2.92	3.86	0.004
	*c* (h^−1^)	0.055	0.002	0.047	0.058	0.021
	*P*_8_ (mmol/g DM incubated)	1.74	0.0478	1.65	1.86	0.024
	*P*_24_ (mmol/g DM incubated)	3.05	0.0725	2.80	3.32	0.008
	*P*_40_ (mmol/g DM incubated)	3.60	0.0788	3.23	3.93	0.006
	*P_max_* (mmol/g DM incubated)	4.02	0.0847	3.52	4.37	0.005
	Final pH	6.21	0.025	6.14	6.32	0.002
Wheat	*a* (mmol/g DM incubated)	0.572	0.0346	0.504	0.657	0.025
	*b* (mmol/g DM incubated)	3.90	0.112	3.38	4.14	0.020
	*c* (h^−1^)	0.047	0.003	0.042	0.067	0.015
	*P*_8_ (mmol/g DM incubated)	1.78	0.0615	1.67	1.85	0.066
	*P*_24_ (mmol/g DM incubated)	3.17	0.0988	2.95	3.31	0.018
	*P*_40_ (mmol/g DM incubated)	3.83	0.106	3.59	4.01	0.020
	*P_max_* (mmol/g DM incubated)	4.47	0.116	4.03	4.70	0.034
	Final pH	6.07	0.037	6.02	6.13	0.004

^1^ SEM: standard error of the mean; Min: minimum; Max: maximum. ^2^
*a*: gas production intercept; *b*: gas production maximum increment; *c*: fractional rate of gas production; *P*_8_: gas production at 8 h of incubation; *P*_24_: gas production at 24 h of incubation; *P*_40_: gas production at 40 h of incubation; *P_max_*: theoretically maximum gas production.

**Table 3 animals-11-01552-t003:** Pearson correlation coefficients *r* between gas production parameters and final pH of rumen in vitro incubations of straw from oats and wheat genotypes from three breeding trials (Oats 1, advanced breeding lines and commercial varieties; Oats 2, commercial and historical varieties; Wheat, advanced breeding lines and commercial varieties).

Parameter Or Variable	*a* (mmol/g DM Incubated) ^1^	*b* (mmol/g DM Incubated)	*c* (h^−1^)	*P*_8_ (mmol/g DM Incubated)	*P*_24_ (mmol/g DM Incubated)	*P*_40_ (mmol/g DM Incubated)	*P_max_* (mmol/g DM Incubated)	Final pH
*a* (mmol/g DM incubated)	1	-	-	-	-	-	-	-
*b* (mmol/g DM incubated)	−0.209 *** ^2^	1	-	-	-	-	-	-
*c* (h^−1^)	−0.363 ***	−0.365 ***	1	-	-	-	-	-
*P*_8_ (mmol/g DM incubated)	0.614 ***	0.169 ***	0.207 ***	1	-	-	-	-
*P*_24_ (mmol/g DM incubated)	0.213 ***	0.607 ***	0.150 ***	0.846 ***	1	-	-	-
*P*_40_ (mmol/g DM incubated)	0.187 ***	0.789 ***	−0.101 *	0.719 ***	0.957 ***	1	-	-
*P_max_* (mmol/g DM incubated)	0.291 ***	0.875 ***	−0.534 ***	0.469 ***	0.702 ***	0.866 ***	1	-
final pH	0.327 ***	−0.567 ***	0.0103 NS	−0.0347 NS	−0.348 ***	−0.425 ***	−0.401 ***	1

^1^*a*: gas production intercept; *b*: gas production maximum increment; *c*: fractional rate of gas production; *P*_8_: gas production at 8 h of incubation; *P*_24_: gas production at 24 h of incubation; *P*_40_: gas production at 40 h of incubation; *P_max_*: theoretical maximum gas production. ^2^ ***: *p* < 0.001; *: 0.01 ≤ *p* < 0.05; NS: non-significant (*p* ≥ 0.10). No tendencies (0.05 ≤ *p* < 0.10) were observed.

**Table 4 animals-11-01552-t004:** Relationships between total gas production at 8, 24, and 40 h of incubation and the theoretically maximum gas production, with proximate composition and morphological variables of straw from three genetic improvement trials (Oats 1, advanced breeding lines and commercial varieties; Oats 2, commercial and historical varieties; Wheat, advanced breeding lines and commercial varieties) incubated in rumen in vitro cultures.

Response	Equation
*P*_8_ (mmol/g DM incubated) ^1^	y=2.17(±0.18; p<0.001)−0.032(±0.0088; p<0.001)ADL−0.012(±0.0042; p=0.006)Cel+0.080(±0.011; p<0.001)avΦ;R2=0.58(p<0.001)
*P*_24_ (mmol/g DM incubated)	y=4.71(±0.30; p<0.001)−0.0748(±0.0142; p<0.001)ADL−0.0239(±0.00678; p<0.001)Cel;R2=0.49(p<0.001)
*P*_40_ (mmol/g DM incubated)	y=6.11(±0.375; p<0.001)−0.103(±0.0178; p<0.001)ADL−0.0371(±0.0085; p<0.001)Cel;R2=0.56(p<0.001)
*P_max_* (mmol/g DM incubated)	y=5.45(±0.162; p<0.001)−0.199(±0.0269; p<0.001)ADL;R2=0.43(p<0.001)
*c* (h^−1^)	y=0.042(±0.0044; p<0.001)+0.00068; p=0.003)ADL−0.0029(±0.00088; p=0.002)RΦ;R2=0.25(p<0.001)

^1^*P*_8_: gas production at 8 h of incubation; *P*_24_: gas production at 24 h of incubation; *P_max_*: theoretical maximal gas production; *c*: fractional rate of gas production; *ADL*: % acid detergent lignin content in the dry matter (DM); *av**Φ*: average diameter of the first and second internodes; *Cel*: cellulose content in the DM; *RΦ*: ratio of diameters of the first and the second internode.

## Data Availability

The data presented in this study are available in Appendix A here.

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
