# Peer review of "Effect of Oats and Wheat Genotype on In Vitro Gas Production Kinetics of Straw"

_animals, 2021, doi:10.3390/ani11061552_

Round 1
Reviewer 1 Report
I appreciate the way the authors have considered the comments from the first round.
Author Response
Thank you for your comments. The Introduction was shortened a bit and the second objective clarified, as per the comments by Reviewer 2. Changes to the text appear in red in the revised version of the manuscript.
Reviewer 2 Report
The paper needs to have a more reinforced introduction going to the point of the topic.
It is not clear to me in fact why the authors take so much in the introduction to get to the objectives. I believe the first part of the introduction is well known to most readers and can easily be skipped. On the other hand, there is no background on the 2nd objective relative to the agronomic conditions. That objective right now is not fully justified even if well discussed. These are my only comments to the manuscript. For the rest, the research is well presented and discussed, but again the "golden thread" needs to start from the introduction.
Author Response
Thank you for your comments. We uploaded a revised version with tracked changes appearing in red. The Introduction has been shortened by eliminating some statements that referred to well known facts and condensing others. With this, the Introduction was shortened from 643 to 495 words (23%). The number of references in the Introduction was also reduced from 27 to 17. Also, the justification of second objective has been clarified (lines 71-74 and 79-80).
Reviewer 3 Report
The paper is very well written. Th authors used adequate methodologies and very well described and discussed their results. In my opinion the paper is acceptable in present form.
Author Response
Thank you for your comments. The Introduction was shortened a bit and the second objective clarified, as per the comments by Reviewer 2. Changes to the text appear in red in the revised version of the manuscript.
This manuscript is a resubmission of an earlier submission. The following is a list of the peer review reports and author responses from that submission.
Round 1
Reviewer 1 Report
The objectives of this work are interesting and well evaluated and described, only some comments.
3. Results. 4. Discussion
Why didn't you complete the analysis of the oats and wheat with the gross energy, fat (or the fatty acid profile) ...?
Why did you not complete the analysis of the gas production of oats and wheat with the determination of methane, volatile fatty acids and total digestibility ...?
I think the work would have been improved.
5. Conclusions
Conclusions should be improved by emphasizing the genotype vs in vitro gas production kinetics correlations, the differences observed between genotypes and in vitro gas production kinetics and in the future and where research could continue in the future...
Author Response
Thank you for your comments, which we found useful to improve the quality of the research presented. Our responses follow, as well as indications of line numbers in the revised version of the manuscript where the requested changes have been made.
REVIEWER: Why didn't you complete the analysis of the oats and wheat with the gross energy, fat (or the fatty acid profile) ...?
AUTHORS: Variation in gross energy and ether extract in straw are minimal, and that is the reason for which they are generally not reported in the proximal analyses (Kernan et al., 1979, 1984; Thomson et al., 1993; Narasimhalu et al., 1998; McCartney et al., 2006; Blümmel et al., 2019; Joshi et al., 2019a; Joshi et al., 2019b). Content of ether extract of winter cereal straws is rather small. According to NRC (Council, 1982), it is on average of 2.2% DM in oats straw and 1.8% DM in wheat straw. There are local results of commercial samples oats and wheat straw containing 0.9 and 0.79% ether extract in the DM (Anrique et al., 2014). With such small content of ether extract we do not think it was worthwhile the complexities and costs of analyzing individual fatty acids in this feeds (and we suspect that much of the ether extract will correspond to waxes rather than fatty acids). Variation in gross energy is mainly dependent on the contents of organic matter and ether extract. Organic matter content varied very little (Table 1 in the manuscript), and ether extract is likely very small. A gross energy analyses would likely find very small coefficients of variation.
REVIEWER: Why did you not complete the analysis of the gas production of oats and wheat with the determination of methane, volatile fatty acids and total digestibility ...?
AUTHORS: We think methane is an important environmental variable, but for this initial screening we wanted to focus on the nutritional characteristics. We are certainly very interested in volatile fatty acids concentration and profile. We took and prepared the corresponding samples and starting analyzing them, but our GC autosampler broke and there is no scheduled date for repair because of the covid-19 pandemic. We conducted apparent digestibility analyses of the pooled straw samples according to Goering and Van Soest (1970), which have been added to Table 1 and also correlated with fermentation variables (lines 347-349). Considerations based on this result have been incorporated into the Discussion in lines 442-445 and 486-490.
REVIEWER: Conclusions should be improved by emphasizing the genotype vs in vitro gas production kinetics correlations, the differences observed between genotypes and in vitro gas production kinetics and in the future and where research could continue in the future...
AUTHORS: We improved the Conclusions recommending comparisons between genotypes contrasting in in vitro gas production kinetics with regards to their digestibility in situ and in vivo (lines 494-496).
References
Anrique, R., Molina, X., Alfaro, M., and Saldaña, R. (2014). "Composición de Alimentos para el Ganado Bovino". Osorno, Chile, Consorcio Lechero: 96.
Blümmel, M., Updahyay, S.R., Gautam, N., Barma, N.C.D., Abdul Hakim, M., Hussain, M. et al. (2019). Comparative assessment of food-fodder traits in a wide range of wheat germplasm for diverse biophysical target domains in South Asia. Field Crops Res. 236: 68-74. doi: https://doi.org/10.1016/j.fcr.2019.03.001
Council, N.R. (1982). United States-Canadian Tables of Feed Composition: Nutritional Data for United States and Canadian Feeds, Third Revision. Washington, DC: The National Academies Press.
Goering, H.K., and Van Soest, P.J. (1970). "Forage fiber analyses (apparatus, reagents, procedures, and some applications)".in: Agriculture Handbook.(ed.) A. R. Service. Washington, D.C., United States Department of Agriculture. 379
Joshi, A., Mishra, V., Chand, R., Chatrath, R., Naik, R., Biradar, S. et al. (2019a). Variations in straw fodder quality and grain-Straw relationships in a mapping population of 287 diverse spring wheat lines. Field Crops Res. 243: 1-7. doi: 10.1016/j.fcr.2019.107627
Joshi, A.K., Barma, N.C.D., Hakim, M.A., Kalappanavar, I.K., Naik, V.R., Biradar, S.S. et al. (2019b). Opportunities for wheat cultivars with superior straw quality traits targeting the semi-arid tropics. Field Crops Res. 231: 51-56. doi: https://doi.org/10.1016/j.fcr.2018.10.015
Kernan, J.A., Coxworth, E.C., Crowle, W.L., and Spurr, D.T. (1979). Straw quality of cereal cultivars before and after treatment with anhydrous ammonia. Can. J. Anim. Sci. 59: 511-517. doi: 10.4141/cjas79-064
Kernan, J.A., Coxworth, E.C., Crowle, W.L., and Spurr, D.T. (1984). The nutritional value of crop residue components from several wheat cultivars grown at different fertilizer levels. Anim. Feed Sci. Tech. 11: 301-311. doi: 10.1016/0377-8401(84)90045-2
McCartney, D.H., Block, H., Dubeski, P., Ohama, A., and Ohama, P. (2006). Review: The composition and availability of straw and chaff from small grain cereals for beef cattle in Western Canada. Can. J. Anim. Sci. 86: 443-455. doi: 10.4141/A05-092
Narasimhalu, P., Kong, D., and Choo, T.M. (1998). Straw yields and nutrients of seventy-five Canadian barley cultivars. Can. J. Anim. Sci. 78: 127-134. doi: 10.4141/a97-020
Thomson, E.F., Herbert, F., and Rihawi, S. (1993). Effect of genotype and simulated rainfall on the morphological characteristics, chemical composition and rumen degradation of the straw fraction of barley plants. Anim. Feed Sci. Tech. 44: 181-208. doi: 10.1016/0377-8401(93)90047-N

Reviewer 2 Report
This is not a good manuscript; it is a simple manuscript of in vitro evaluation of two feed ingredient. It has not any novelty and a lot f papers were already published with the same idea and objective, sorry this paper does not reach t the stander to publish in a high quality journal such as Animals, So, i have to reject it.
Author Response
REVIEWER: it is a simple manuscript of in vitro evaluation of two feed ingredient
AUTHORS: We think that unfortunately the Reviewer missed the main hypotheses. The manuscript is not a descriptive presentation of results of two feed ingredients (oats and wheat straw) evaluated in vitro but an evaluation of differences among genotypes in in vitro gas production kinetics within each type of feed, and their association with important agronomic traits such as grain yield and the incidence of lodging and diseases. In addition the research explored the associations between gas production kinetics and proximate composition.
Reviewer 3 Report
This manuscript is about comprehensive work on in vitro gas production of cereal straw. Overall, the manuscript is well written, the study appears thoroughly conducted, and methods well presented, including statistics. Results presentation is too excessive in regard to non-existing effects or associations.
The results were predictable to a fair extent, keeping in mind the principle relationships between fibre fractions, lignin, and microbial activity in the rumen. However, with the focus on a systematic investigation of genotype differences the paper has some novel character.
It is unfortunate that straw from the plots was pooled into one sample per genotype for the in vitro study. Real field replicates are lacking. Hence, any estimate of genetic parameters is not possible. While differences between genotypes occurred, it may be over-ambitious to conclude that the differences indicate a potential for breeding.
Specific remarks:
Simple summary is basically identical with the abstract. It should be a simple summary.
38 should be mentioned that the trials were conducted in Chile
43 please double-check whether the last sentence of the abstract really is substantiated by your data (see comment above on non-existing field replicates)
101 does this mean approval was obtained long after the experiment was done? Or was an approval not necessary for this specific study because cannulated animals and their use as donor animals was approved before and in a more general way?
140 this is unclear. Were all straw in one experiment samples the same day? Or different days within the one week? Please be more specific about standardisation of procedure.
178 sure the cows were only fed the straw and had access to the mineral block but without an additional nitrogen source?
221 delete one ‘pressure’
Table 6 may not be necessary. Only very few of the coefficients were statistically significant. The significant could be mentioned in the text only.
Why are the gas production parameters in this table expresses in atm and not converted to mmol such as in the other tables?
Fig. 2 may not be necessary. None of the displayed relationships was significant, a fact that can be mentioned in the text only.
471: price is not given in %
Author Response
Thank you for your comments, which have been helpful to improve our manuscript. Please find our responses below along with indications of the line numbers where changes have been incorporated.
REVIEWER: Results presentation is too excessive in regard to non-existing effects or associations.
AUTHORS: In agreement with your recommendations we have moved Table 6 in the original version of the manuscript to Supplementary materials (now Supplementary Table 2 in the revised version). Figure 2 was eliminated, and Table 2 in the original version of the manuscript was also moved to Supplementary materials (now Supplementary Table 1 in the revised version).
REVIEWER: It is unfortunate that straw from the plots was pooled into one sample per genotype for the in vitro study. Real field replicates are lacking. Hence, any estimate of genetic parameters is not possible. While differences between genotypes occurred, it may be over-ambitious to conclude that the differences indicate a potential for breeding.
AUTHORS: Previous to conducting the study, we did discuss the possibility of incubating each sample of straw collected in the field as an experimental unit. However, incubating 100 different samples (25 different genotypes x 4 blocks) with 2 replicates (considering that 3 replicates would have actually been ideal (Yáñez-Ruiz et al., 2016)) would have meant incubating 200 bottles per run, which exceeds our capacity (and probably the capacity of most laboratories) as more bottles mean longer time between the first and last batch are inoculated, resulting in differences in inoculum viability. We also gave some thought to the possibility of incubating each of the field blocks in a different incubation run, but then we would have had a confounded effect between field block and incubation run.
We are thinking in using the results of this initial screening to identify a few contrasting genotypes and grow them in the field, and collect them and incubate each plot separately, so as to assess variation within each genotype with field replicates.
REVIEWER: Simple summary is basically identical with the abstract. It should be a simple summary.
AUTHORS: The Simple summary has been revised and rewritten so at to be understandable by a general, non-scientific readership (lines 17-29).
REVIEWER: 38 should be mentioned that the trials were conducted in Chile
AUTHORS: Clarification made (line 37).
REVIEWER: 43 please double-check whether the last sentence of the abstract really is substantiated by your data (see comment above on non-existing field replicates)
AUTHORS: We agree that this conclusion is overreaching. It has been replaced by a recommendation to confirm in vitro results with animal experiments (lines 41-42). The discussion about the potential for breeding has also been removed from the Discussion and Conclusion sections.
REVIEWER: 101 does this mean approval was obtained long after the experiment was done? Or was an approval not necessary for this specific study because cannulated animals and their use as donor animals was approved before and in a more general way?
AUTHORS: The cannulated animals had been used in other projects and their cannulation had received previous approval. The research presented in this manuscript was internally funded and our administration had not requested an approval from the institutional bioethics committee to conduct the experiments. So we applied for the institutional bioethics committee´s approval after the experiments were conducted. Admittedly, it was an atypical procedure.
REVIEWER: 140 this is unclear. Were all straw in one experiment samples the same day? Or different days within the one week? Please be more specific about standardisation of procedure.
AUTHORS: We meant that straw was not left in the field for more than one week after the grain was harvested. All the straw samples of each breeding trial (Oats 1, Oats 2 and Wheat) were sampled within one day, but each of the breeding trials was harvested and sampled on different days. Clarified in lines 138-141.
REVIEWER: 178 sure the cows were only fed the straw and had access to the mineral block but without an additional nitrogen source?
AUTHORS: We did not provide the cows any protein supplement but monitored their body condition and general health status when feeding and sampling rumen contents. Whenever they visibly lost condition they were removed from the diet and the experiment for about a month and grazed good quality pastures with the research farm´s herd, until they regained body condition, and then they were returned to the straw diet. They were adapted to the straw for two weeks before being sampled again.
REVIEWER: 221 delete one ‘pressure’
AUTHORS: First mention of pressure deleted (line 224).
REVIEWER: Table 6 may not be necessary. Only very few of the coefficients were statistically significant. The significant could be mentioned in the text only.
AUTHORS: Table 6 has been moved to Supplementary materials as Supplementary Table 2 in the revised version.
REVIEWER: Why are the gas production parameters in this table expresses in atm and not converted to mmol such as in the other tables?
AUTHORS: This was a mistake in the figure, which does not affect the coefficients of determination because pressure and number of moles are linearly related (it would only affect the slope of the regressions, which was not provided in the table). The mistake has been corrected in Supplementary Table 2 in the revised version and gas production appears in mmol.
REVIEWER: Fig. 2 may not be necessary. None of the displayed relationships was significant, a fact that can be mentioned in the text only.
AUTHORS: Figure 2 has been deleted, as recommended.
REVIEWER: 471: price is not given in %
AUTHORS: The sentence was rewritten for clarity (lines 485-486).
References
Yáñez-Ruiz, D.R., Bannink, A., Dijkstra, J., Kebreab, E., Morgavi, D.P., O’Kiely, P. et al. (2016). Design, implementation and interpretation of in vitro batch culture experiments to assess enteric methane mitigation in ruminants—a review. Anim. Feed Sci. Tech. 216: 1-18. doi: https://doi.org/10.1016/j.anifeedsci.2016.03.016
